# Increased Temperature Effects During Fruit Growth and Maturation on the Fruit Quality, Sensory and Antioxidant Properties of Raspberry (*Rubus idaeus* L.) cv. Heritage

**DOI:** 10.3390/foods14071201

**Published:** 2025-03-29

**Authors:** Francisca Aguilar, Martina Salazar, Lida Fuentes, Daniel Calderini, Alejandro Jerez, Carolina Contreras

**Affiliations:** 1Instituto de Producción y Sanidad Vegetal (IPSV), Facultad de Ciencias Agrarias y Alimentarias, Universidad Austral de Chile, Valdivia 5090000, Chile; francisca.aguilar@alumnos.uach.cl (F.A.); danielcalderini@uach.cl (D.C.); 2Centro Regional de Estudios en Alimentos Saludables (CREAS), Avenida Universidad 330, Valparaíso 2373223, Chile; msalazar@creas.cl (M.S.); lfuentes@creas.cl (L.F.); 3Pontificia Universidad Católica de Valparaíso, Avenida Brasil 2950, Valparaíso 2340025, Chile; 4Laboratorio de Farmacia (Instrumentación Analítica), Instituto de Farmacia, Facultad de Ciencias, Universidad Austral de Chile, Valdivia 5090000, Chile; alejandrojerez@uach.cl

**Keywords:** climate change, heating treatments, sensory panel, anthocyanins, polyphenolic compounds

## Abstract

Red raspberries are valued for their color, flavor, and health-promoting organic compounds, which may be affected by increased temperatures due to climate change. This work aimed to provide new information on the response of raspberry cv. “Heritage” to thermal increase and its impact on fruit quality and perceived flavor. The study was conducted during two seasons in two locations with contrasting agroclimatic conditions. A zone with high background temperatures (central orchard) and low background temperatures (southern orchard) were studied. The treatments were three heating chambers installed at the fruit set, increasing the ambient temperature by ~4 °C, and untreated controls. Heat-treated raspberries were larger than the controls but showed softer fruit. Soluble solids were lower in treated fruit, while titratable acidity did not show a consistent pattern between treatments or orchards. Flavonoid content and anthocyanins did not present a clear pattern for both orchards and seasons. Heated raspberries had lower vitamin C in both years and orchards. The sensory analysis revealed differences only in color uniformity in the heated fruit from the central zone. Increased temperatures will affect the quality and nutritional aspects of the raspberries; however, at a sensory level, these changes may not be perceived.

## 1. Introduction

Consumers’ interest in maintaining a healthy diet has risen in recent years, influenced by better access to information about the relationship between diet and health, sociodemographic profiles, and knowledge about nutritional aspects [1,2]. Worldwide, raspberry production reached 852,000 tons in the 2023/2024 season. Chile produced 16,083 tons in 2024, with 74% destined for the frozen market, while the remaining 26% was destined for markets that included fresh, dehydrated, and local consumption [3,4].

Red raspberry (*Rubus ideaus* L.) is appreciated for its attractive color, unique flavor, aroma, and various organic compounds beneficial to health. These compounds involve a significant source of vitamins, minerals, dietary fiber, and antioxidants, with a phytochemical profile rich in anthocyanins [5,6,7]. Metabolite composition, including metabolites affecting taste (sugars, acids), flavor (volatiles), and health benefits (antioxidants), can be affected by variety, maturity, growth, and environmental conditions [8,9,10,11,12]. A key climate variable affecting fruit production and quality traits is temperature, which is increasing in the present climate change conditions [13]. Several studies have examined the impact of climate change on crops, focusing primarily on grain crops like rice, corn, and wheat, as well as some legumes such as soybeans, due to their significant contribution to the human diet [14,15,16]. However, there is a lack of research on the effects of climate change on fruit crops, which hinders the necessary understanding to address this challenge [17]. Red raspberry is a species resistant to cold and drought but susceptible to high temperatures, a factor limiting the optimal production of this crop [18]. In this regard, understanding the increasing weather temperatures on raspberry fruit is a requirement to develop breeding and management strategies to deal with climate change and adapt fruit production and quality.

The fruit quality is largely influenced by preharvest growing conditions, especially temperature. According to Osatuke and Pritts [19], the soluble solids content (SSC) and titratable acidity (TA) of strawberries were positively correlated with the air temperature differential during fruit ripening, and there was no significant difference in fruit quality attributes such as SSC, TA, aromatic volatile concentration, and polyphenolic compounds by variations in soil carbon inputs, nitrogen rates, pesticides, and microbial supplements. Their study suggests that growers should prioritize temperature management over agricultural inputs to impact the SSC and TA of strawberries. Heat stress in red raspberries is evidenced by smaller berries, lower yields, and poor fruit quality, especially when summer temperatures exceed 32 °C [20]. Furthermore, higher-than-optimal temperatures have been shown to reduce anthocyanin content in various fruit species [20,21]. In another study, raspberry fruit from a protected soilless culture, with a wider range of temperature and relative humidity, showed higher weight but less soluble solids content than fruit from an open field [22]. The same study showed that fruit from the open field had a higher content of sugars such as D-fructose, and the sensory panel showed significant differences in the perception of sweetness, acidity, color, and firmness of ripe fruit from both growing conditions.

Another less-studied consequence of increased temperatures during the growing season is nutrient dilution, which will directly impact the global population’s health. Research on the nutritional composition of staple crops has found a declining trend in nutrients due to factors such as increases in CO_2_ and temperature, causing deficiency in micronutrients, proteins, and vitamins, among others [23,24,25]. For example, the authors of [26] observed a negative effect of climate change on nitrogen use efficiency in wheat. In rice, a consistent decline in B vitamins, N-based vitamins, and proteins was reported [27]. In fruits, a recent study on wild blueberries reported that total soluble protein decreased, among other quality traits, concluding that the nutritional value of blueberries may be reduced with increased environmental temperatures [28].

Therefore, this work aimed to evaluate the impact of increased temperatures on red raspberries caused by climate change and to study their effect on raspberry quality traits, including antioxidant and sensory attributes. This study provides valuable preliminary information and a more comprehensive insight into the future raspberry response to environmental temperature increment.

## 2. Materials and Methods

### 2.1. Plant Materials

Raspberry cultivar “Heritage” from two commercial orchards was studied during two seasons, 2023 and 2024. Both orchards were located in two contrasting agroclimatic regions of Chile 944 km apart. The first orchard was located in Central Chile with a high background temperature zone, in Casablanca (33°20′45″ S 71°22′10″ W), Valparaíso Region. The second orchard was located in Southern Chile, in a low background temperature zone, in Paillaco (40°07′06″ S 72°52′16″ W), Los Ríos Region.

The central orchard was 7 years old, and the raspberry canes were pruned yearly in July (winter). On the other hand, the southern orchard was 8 years old, and the canes were pruned in August (winter) and thinned in January (spring), two months before harvest. The raspberry plants were approximately 90 cm high and 50 cm apart on the rows. The two orchards were handled with similar standard commercial practices, which included drip irrigation systems and fertilization analyses (Appendix A).

### 2.2. Treatments

Two temperature treatments were carried out across locations and years, with a control treatment left at ambient temperature and an increased temperature treatment by ~4 °C over the ambient. Three heating chambers were randomly installed in each orchard, and each chamber represented a replicate (Figure 1). Each chamber was 6 m × 2 m × 2.1 m (L × W × H), made of a wooden frame, wrapped with a high-density transparent polyethylene plastic film of 100 µm (Delsantek S.A., Santiago, Chile), and equipped with electric heaters according to the methodology used by Lizana et al. [29]. Temperatures and relative humidity inside and outside of each chamber were monitored hourly with HOBO MX2301 sensors (Bourne, MA, USA), each sensor was placed 80 cm above the ground. In addition, PAR radiation data were collected from the Casablanca (Agromet, National Agroclimatic Network) weather station located 16.5 km from the central orchard, and the Paillaco (Agromet, National Agroclimatic Network) experimental station located 20 km from the southern orchard.

In both years, the heating chambers were installed at fruit set and closed when the fruit was at the phenological green stage. When chambers were closed, all raspberry fruits with advanced developmental stages, such as white, pink, orange, or red were removed, and only green stage fruit were kept. Chambers had a folding curtain that was opened from 9 a.m. to 6 p.m. during the day to prevent excessive temperature rise and was closed at night. The heating treatments were imposed for 12 days in the central orchard and for 11 days in the southern orchard.

### 2.3. Harvest

Raspberries were harvested at the red ripening stage as reported by Álvarez et al. [30], from sound plants with similar vigor. The fruit was hand-picked between 7 and 8 a.m. to avoid excessive heat and solar exposure to the fruit. The central orchard was harvested on 22 February 2023, and the southern orchard was harvested a month later, on 22 March 2023. For both orchards, fruit was picked from the upper third of the canes, that is, primocanes or one-year-old shoots at the end of the summer season, according to García et al. [31]. In the second season, fruits were harvested for the central orchard on 26 February 2024 and on 21 March 2024, for the southern orchard.

Then, 60 g was harvested for quality assessments, 50 g for antioxidant analyses, 50 g for vitamin C analysis, and 140 g from each heating chamber and control. Additionally, for sensory analysis, an extra ~450 g of fruit was collected per control. For each year and each orchard, harvested fruit was placed in 250 g clamshells and immediately transported in a cooler equipped with refrigerant gel units, maintaining the temperature between 3 and 4 °C. Raspberries from the central orchard were transported to the Centro Regional de Estudios en Alimentos Saludables (CREAS), Valparaíso, located 31.4 km from the orchard. Samples from the southern orchard were analyzed at the Fruit Postharvest Laboratory of the Universidad Austral de Chile, 64.4 km away.

Once in the laboratory, samples for antioxidant analysis and vitamins were stored in bags and placed at −20 °C for further analysis. Fruits harvested for the sensory panel were stored at 4 °C for 2 days, simulating commercialization time for the fresh market. Later, raspberries were transferred to the Laboratory for Quality Assurance of Measurement/Analytical Division facilities at Universidad Austral de Chile.

### 2.4. Fruit Quality Assessments

Twenty raspberry fruits per treatment were used for quality assessments. Firmness was determined as the force needed to deform the fruit in 1 mm with a 25 mm diameter probe using the FirmPro equipment (Happy Volt SPA, Santiago, Chile) [22] and expressed in gF/mm. Later, polar and equatorial diameters were measured using a digital caliper (Kawasaki, Japan). Fruit weight was determined with an Ecobeck digital balance, model SF-803 (Jiangyin, China). Color was assessed with a CR-400 tristimulus Minolta colorimeter (Ramsey, NJ, USA), and values obtained were expressed as brightness (L*), chroma or intensity (C*), and hue (°). The soluble solids concentration (SSC) was evaluated using an ATAGO PAL-1 digital refractometer (Tokyo, Japan) and expressed as a percentage of °Brix. Titratable acidity (TA) was measured in 2 mL of juice, adding 8 mL of fresh distilled water, and then titrating with a 0.1 N NaOH solution until a pH of 8.2 was reached using a digital pH meter (Hanna instrument, Woonsocket, RI, USA). Results were expressed as percentage of citric acid.

### 2.5. Analyses of Antioxidant Capacity Related to Polyphenolic Compounds

#### 2.5.1. Extraction of Polyphenolic Compounds

The extractions were carried out according to Zuñiga et al. [32], with 2 g of sample ground in liquid nitrogen, which were homogenized with 40 mL AWA (acetone/water/acetic acid, 70:29.5:0.5 in *v*:*v*:*v*), then this mixture was incubated at 30 °C for 40 min with Vortex agitation for 30 s every 10 min. Subsequently, it was centrifuged at 5000 rpm or 4050× *g* for 10 min at a temperature of 15 °C. The supernatant was passed through 125 mm filter paper with 20–25 µM pores, code 1238. The extracted volume was measured, and the samples were stored at −20 °C. These extractions were used to measure total flavonoids, total polyphenols content (TPC), and antioxidant capacity by ORAC [33]. All chemical reagents for the procedure above and the following ORAC, TPC, flavonoid, and anthocyanin contents were purchased from Merck Company (Darmstadt, Germany).

#### 2.5.2. Antioxidant Capacity (ORAC)

The ORAC activity was measured using a 50-fold diluted AWA extract according to Valdenegro et al. [33] with modifications. Twenty-five microliters of sample previously diluted in PBS (phosphate-buffered saline) and 150 µL of fluorescein [108 nM] were added, then the mix was incubated for 30 min at 37 °C and shaken every 3 min. Then, 25 µL of APPH (2,2′-Azobis(2-amidinopropane) dihydrochloride) [153 mM] were added to the samples in the black 96-well plate and placed in a multimode microplate reader (Fluoroskan Ascent, Thermo Scientific; Waltham, MA, USA) and incubated at 37 °C for 60 min, shaking every 3 min. Fluorescence was monitored every 3 min throughout the experiment. Each sample was analyzed in triplicate. ORAC activity results were estimated based on a Trolox standard curve using a quadratic regression equation obtained from Trolox concentration and the net area under the fluorescence decay curve. ORAC activity was expressed as mmol TE/g FW.

#### 2.5.3. Total Polyphenol Content (TPC)

The TPC protocol was determined using the method proposed by Singleton and Rossi [34]. First, the sample (0.5 mL) was diluted with 3.75 mL water, and 0.25 mL of Folin–Ciocalteu reagent diluted to 50% in aqueous solution was added; then, 0.5 mL of 10% sodium carbonate was added and the mixture was homogenized. Subsequently, the mixture was left to stand for 1 h at room temperature away from light. Two blanks were made by the same procedure, replacing the sample with the AWA extraction solution. The absorbance of the samples was determined at 765 nm in a UV/Visible spectrophotometer. The results were expressed as mg gallic acid equivalent (GAE)/g FW.

#### 2.5.4. Total Flavonoid Content (TFC)

The TFC was measured using the method proposed by Chang et al. [35]. Five hundred of the AWA extract was mixed with 1.5 mL of 95% ethanol, 100 µL of 10% aluminum chloride AlCl_3_, 100 µL of 1 M potassium acetate CH_3_CO_2_K, and 2.8 mL of distilled water. After incubation for 30 min at room temperature, the optical density (OD) of the reaction mixture was determined at 415 nm in a UV/Vis spectrophotometer (model UV-160A, Shimadzu; Kyoto, Japan), using a standard curve prepared with quercetin-3-glucoside at concentrations 0.0125–0.15 mg/mL. The results were expressed as mg quercetin equivalent (QE)/g FW.

#### 2.5.5. Total Anthocyanin Content (AC)

The AC was quantified by the pH differential method [36,37] with some modifications [36]. Raspberry fruits (2.5 g) were homogenized using 10 mL of absolute ethanol with 1% HCl (85:15 *v*/*v*) as an extraction solution, incubated overnight at 4 °C, and centrifuged for 15 min at 8000 rpm or 10,370× *g* at 4 °C. Two aliquots from the ethanolic phase of each sample were diluted (1:4) with two different buffers: a pH 1 buffer (0.025 M KCl) and a pH 4.5 buffer (0.4 M sodium acetate). Finally, absorbances were quantified at 516 and 700 nm. Total anthocyanin content was calculated based on the Lambert–Beer law, using the coefficient of molar extinction for the cyanidin-3-glucoside (34,300 M^−1^ cm^−1^) reported by Siegelman and Hendrick [38]. Results from each treatment (three biological and two technical replicates) were expressed as mg of cyanidin-3-glucoside equivalent (C3G)/g FW.

### 2.6. Vitamin C Analysis

The analytical methodology by HPLC for the quantification of ascorbic acid in raspberry samples was adapted from the work of Van de Velde et al. [39]. Approximately 10 g of raspberry frozen samples were crushed for chromatographic analysis. For the extraction, 5 g of homogenized raspberries were added to 25 mL of extracting solution (orthophosphoric acid (30 g/L) and acetic acid (80 g/L)). The mixture was homogenized for 1 min, sonicated in an ultrasonic bath for 15 min, and then centrifuged at 10,000 rpm or 10,000× *g* for 20 min at 4 °C. The supernatant was separated, and 1 mL was diluted with mobile phase to achieve a final volume of 6 mL, then filtered through a 0.45 μm Millipore membrane and injected in the HPLC system for quantifying the content of ascorbic acid (vitamin C). For quantification, 1 mL of supernatant was added with 0.2 mL of _DL_-dithiothreitol, DTT solution (5 mg/L DTT prepared in 2.58 M potassium phosphate dibasic) [39,40].

The raspberry extracts were analyzed in an HPLC system, which was a Shimadzu Prominence instrument equipped with a quaternary pump system and DAD detector (Shimadzu, Kyoto, Japan), and the software used for the analysis was LabSolutions^®^ (Version 5.106, Shimadzu Corporation, Kyoto, Japan). The chromatographic separation was carried out using a Kromasil C18 column with dimensions of 4.6 × 250 mm and a particle size of 5 µm width. A 266 nm amount was the selected wavelength for the detection of ascorbic acid. The runtime was set for 7 min to ensure complete elution of compounds in the samples. The volume injected was 10 µL and the temperature of the column was set at 25 °C. The mobile phase, under isocratic conditions, consisted of a 95% (0.03 M) sodium acetate/acetic acid buffer and 5% methanol, and the pH of the mobile phase was 5.8 with a 1.0 mL/min flow rate.

A standard solution prepared with orthophosphoric acid (30 g/L) and acetic acid (glacial) (80 g/L) was used for quantification. Adaptations to the work of Van de Velde et al. [39] and the recommendations of the Association of Official Analytical Chemists (AOAC) [41] were considered. A stock standard solution of ascorbic acid (4.0 g/L) was prepared in an extracting solution of orthophosphoric acid (30 g/L) and acetic acid (80 g/L). For doing this solution, 0.1 g of ascorbic acid powder, oven-dried for 1 h at 105 °C, was dissolved in 25 mL of the extracting solution, protected from light, and stored at 4 °C until use. From this solution, different concentrations were prepared to obtain a calibration curve that covered the range of 10 to 100 mg/L. All chemical reagents were purchased from Merck Company (Darmstadt, Germany), with the exception of L-ascorbic acid, which was purchased from Sigma-Aldrich company (Burlington, MA, USA).

### 2.7. Sensory Analyses

The “Difference-from-Control” sensory test [42,43] was chosen to evaluate raspberries from both orchards, in order to identify possible differences in sensory attributes between fruit harvested from heat-treated or untreated control plants.

Ten panelists, highly trained in different food products, participated in a two-hour orientation session to select specific attributes describing raspberry fruit quality [44]. On the day of the sensory test, the raspberries were taken out of the 4 °C cooler at 7 a.m. to increase the flesh temperature up to 18–20 °C until the time of the test at 10 a.m. Fruit evaluation was performed in sensory booths separated by a melamine wall which provides odor isolation and privacy for each panelist. The room was maintained at an ambient temperature of 20 °C with artificial cool-toned lighting. Three to four raspberries were presented per cup. Each cup was an odorless, transparent plastic 60 mL cup with a lid. Panelists were presented with two pairs of cups, one cup labeled as “reference” (unheated control), and one cup labeled with a 3-digit code. The coded sample was either the berries from the heated plants or berries from the control. Panelists repeated the test in 3 sessions. All sessions were performed on the same day, with a 15 min break between sessions. Panelists evaluated 4 quality aspects in the following order: aroma, appearance, texture, and flavor. Panelists rated the size of the difference between the reference and the coded sample by marking a line scale from −50 to +50, with zero indicating “same as reference”. The scale was anchored at the extremes with the following words: overall aroma (less intense −50/more intense +50); aroma of ripe/unripe berry (unripe −50/overripe +50); off aroma such as chemical or moldy (less off-odor −50/more off-odor); color intensity (duller −50/brighter +50); color uniformity (less uniform −50/more uniform +50); uniformity of drupelets (heterogeneous −50/homogeneous +50); firmness (druplets not cohesive −50/cohesive druplets +50); juiciness (dry −50/juicy +50); overall raspberry flavor (less flavor −50/more flavor +50); off-flavor such as chemical of moldy (less off-flavor −50/more off-flavor +50); sweetness (less sweet −50/sweeter +50) and sourness (less sour −50/more sour +50). Data were recorded and statistically analyzed using Compusense^®^ (Version 23.0.25636 2024/01/09, Compusense Inc., Guelph, ON, Canada).

### 2.8. Statistical Analysis

Three biological samples were used per treatment (heating and control treatments) for each analysis (quality, antioxidant, vitamin C, and sensory attributes). In the case of antioxidant analyses, two additional technical replicates were carried out.

Statistical analysis of fruit quality and sensory analysis data was carried out using a two-factor statistical model with interaction. The factors considered were location and temperature, which influenced the quality parameters and sensory attributes of the raspberries. The Tukey multiple comparison test was performed to determine statistical differences after the ANOVA with a *p*-value ≤ 0.05 for quality parameters, and Fisher’s Least Significant Difference at a *p* ≤ 0.05 for sensory analysis.

The statistical model applied to quality and nutritional (antioxidant capacity, total polyphenols, total flavonoids, total anthocyanin, and vitamin C) data, was a bifactor model with interaction, considering the factors of location and temperature. An ANOVA was performed, and LSD multiple comparison test with *p*-value ≤ 0.05 was carried out. This test was selected due to the low variability of the data; therefore, a more sensitive test was required to identify significant differences detected in the ANOVA. All data were statistically analyzed using RStudio (Version 2024.12.1 + 563, Boston, MA, USA).

## 3. Results

### 3.1. Weather Conditions and Temperature Regimes

Higher background temperatures were recorded in the central orchard during the growing cycle/treatment in a period of 12 days (10–22 February) in the 2023 season, than in the southern orchard with a treatment period of 13 days (9–22 March) (Table 1). In 2024, the same temperature pattern as in 2023 was observed where the central orchard presented higher background temperatures; the treatment at the central orchard was for 12 days (14–26 February) and 10 days (11–21 March) for the southern orchard (Table 1). In the first experimental year, the temperatures inside the chambers increased by 1.5 °C in the central orchard and 4.39 °C in the southern one (Table 1). For the second experimental year, temperatures increased by 3.06 °C in the central orchard and 3.95 °C in the southern orchard.

In regard to the radiation, during the 2023 year, the central orchard registered an average maximum radiation/hour of 282.39 W/m^2^, and the southern orchard 147.33 W/m^2^ during the days of the experiment. Likewise, in 2024 the central orchard had an average maximum radiation/hour of 255.75 W/m^2^ and the southern orchard 212.05 W/m^2^.

### 3.2. Effect of Higher Temperature on Fruit Quality Traits

The fruit collected from the heating chambers in both years was larger than the controls, as indicated by an increase in fruit weight (Figure 2). For the 2023 season, the heating treatment increased fruit weight by 8.3% in the central orchard, while raspberries from the southern orchard increased fruit weight by 23%. In 2024, the heating treatments increased raspberry fruit weight by 28% and 13% in the central and southern orchards, respectively (Figure 2). In a detailed view, during the 2023 season, the central orchard had average weights of 2.3 g and 1.8 g for the heat treatment and control, respectively. On the other hand, for the southern orchard, the average weights achieved were 2.3 g and 2.0 g for the heat treatment and control, respectively. In the 2024 season, for the central orchard, the average weights achieved were 2.5 g and 2.3 g for heat treatment and control, respectively, while in the southern orchard, the average weights were 2.9 g and 2.3 g for heat treatment and control, respectively. In contrast, fruit firmness had the opposite pattern compared to weight. Fruits harvested from the heating chambers were softer than the control treatment in both locations and during the two seasons. The loss of firmness was higher in the central orchard than the southern, with a drop in firmness of 8 and 3 gF mm^−1^, respectively, for the 2023 season. This fruit firmness decline was less evident in 2024, and the southern orchard showed no significant differences between heated and control fruit. As for SSC, heated fruit presented lower sugar accumulation than the control fruit, consistently in both locations and years. On the other hand, TA was not consistent between the two seasons, where 2023 year had no significant differences between treatments and orchards, and 2024 only showed differences in the central orchard in which acidity decreased by 19% (Figure 2).

The effect of orchard and heating treatment did not affect fruit weight independently, but rather both factors seemed to have a synergistic effect (Table 2). Interestingly, for firmness, both factors were highly significant during the 2023 and 2024 years, but the interaction was not significant, indicating an independent effect of each factor. The results for SSC and TA were similar to those observed for fruit weight in both seasons, that is, the interaction between the orchard and the heating treatment was significant.

### 3.3. Effect of Higher Temperature on Antioxidant Properties

Fruit collected from the southern orchard had a higher ORAC than fruit collected from the central orchard, and a higher antioxidant capacity was observed in 2023, for both locations, compared to 2024 (Figure 3). On the other hand, the results for TPC were not consistent in both seasons: in 2023 the fruit with the highest TPC (2.12 ± 0.19 mg of GAE/g FW) was that from the central orchard, while in 2024, the fruit collected from the southern orchard had the highest TPC (2.13 ± 0.16 mg of GAE/g FW). Figure 3 shows that, in general, the fruit harvested from the heating chambers in both orchards for both seasons decreased by 13.1% in total phenol content. For the flavonoid content, there were no differences between treatments in the first season, whereas the behavior was the opposite for both orchards in the second season. In the central orchard, the raspberries from the control reached a higher flavonoid content (0.10 ± 0.02 mg of quercetin/g FW), while in the southern orchard, raspberries from the heating chambers reached a higher flavonoid content (0.19 ± 0.06 mg of quercetin/g FW). Anthocyanins followed a similar pattern to the total phenol content (TPC): in the first season, anthocyanins decreased in fruit from the heated plants in comparison with the control. In the second season, the raspberries harvested from the heating chamber of the southern orchard reached a higher content (0.31 ± 0.04 mg of C3G/g FW) (Figure 3).

The statistical analysis for each of the antioxidant parameters and polyphenolic compounds content (Table 3) showed that for the antioxidant capacity, the individual factor that had an effect in both seasons was the orchard, while for the second season, both individual factors plus the interaction of these had a significant effect on the response variable. The total polyphenols content (TPC) was significant for the individual factors corresponding to orchard location and heat treatment; however, the interaction of these two factors was only significant in 2024. The flavonoid content was significant only for the interaction of the two study factors in both seasons, which means that the combination of levels of each factor affected the variable in question. Finally, the factor that had a consistent effect in both seasons was the heat treatment, while the orchard only had an effect in the second season (Table 3).

### 3.4. Vitamin C

The average values of vitamin C or ascorbic acid content found in the heat-treated and the control fruit showed a sharp decline in the heated fruit in both orchards and seasons (Table 4). Interestingly, all values of vitamin C were lower in the southern orchard compared to the central orchard regardless of the treatment.

Table 5 shows the statistical significance of the fruit collected from both orchards in the two seasons. In the first season, 2023, despite the fact that a decrease in vitamin C was observed in all heat-treated raspberries, no statistical significance for the factors studied (orchard or heat treatment) was found. However, for the second season, 2024, each factor independently showed a significant effect, but no interaction between them was found.

### 3.5. Sensory Analyses

Sensory evaluation showed no difference between heat-treated fruit and control in both orchards, except for fruit color uniformity in the central orchard (Figure 4). Fruit from the heat-treated plants had less color uniformity (more variable) than fruit from the unheated control plants. Drupelet size was also less uniform in the fruit from heat-treated plants than from untreated control, even though the difference was not significant. Fruit-to-fruit variability was likely too large for panelists to perceive a difference between fruit from heat and unheated control, as can be seen with large error bars and a difference of up to −5.0 (on a −50 to +50 linear scale) could be observed between coded control and reference control fruit for sourness (Figure 4). Another explanation for the lack of differences is the fact that too few fruits were presented at each session.

## 4. Discussion

### 4.1. Fruit Quality

Among the affected quality parameters was fruit weight. It was observed that there was an effect of orchard location since fruit harvested from the southern orchard had similar or higher weight than fruit collected from the central orchard for both seasons. Likewise, higher average weights were obtained in heat-treated raspberries. Similar results were observed in a previous study conducted in Central Chile on raspberries grown under protected soilless (greenhouse) and open field conditions [22], the average fruit weights were 3.9 ± 0.4 g for the protected soilless condition, and 2.6 ± 0.2 g for the open field coinciding with our results where weights were higher under increased temperature conditions. The dynamic observed in the weight of the raspberries is attributed to the thermal increase caused by the heating chambers, which generated optimal internal conditions for growth in terms of fruit quality. Other studies in which a similar heating system was used describe a physiological effect on the plants by generating a greater absorption of water and nutrients under these higher temperature environments and, therefore, higher photosynthesis rates, which may explain the increase in fruit growth [45,46]. The optimal growth temperature reported for raspberries is 18 °C, as temperatures above 25 °C reduce photosynthesis and growth [47]. Although much higher temperatures were reached in our study under the heating chambers, weights were not reduced, probably due to the timing in which the increased temperatures were imposed. The heat treatments were applied after the fruit set, and to trigger a weight shift, it should have been implemented earlier in the cane development, i.e., after the bud set. Another possible explanation for the observed increase in weight, instead of a decrease, might be a specific interaction between temperature and carbon supply during berry growth. It has been reported that the dominant effect was the temperature since temperature intensity and threshold of 42–45 °C were established as a determinant of lowering the berry weights of grapevine cv. Shiraz [48]. The heating treatment in this study also indicates that the temperature is the dominant factor; however, other external effects or soil characteristics may also condition the fruit response.

Fruit softening is a process associated with ripening accompanied by a loss of firmness caused by changes in cell wall structure and composition [49,50]. Abiotic stress, such as high temperature, affects the fruit texture, leading to changes in the chemical composition of the cell wall and reducing the calcium involved in fruit firmness and, therefore, increasing softening [13,51]. In our study, in both seasons, a lower firmness was observed in fruit with the heat treatment, suggesting that under future climate change conditions, fruit may be softer with shorter postharvest life.

Several authors have examined the impact of climate change in SSC and TA on important fruit species, such as grapevines and, recently, blueberries. In grapevines, four heat treatments (day heating, night heating, day/night heating, and untreated control) were evaluated in *Vitis vinifera* L. cv. “Shiraz”; the total soluble solids and berry weight were significantly lower under high day temperatures [48]. Likewise, Biasi et al. [52] evaluated the climate change temperatures on different grapevine varieties during a period spanning from 1995 to 2015. The results revealed that soluble solids in varieties such as “Cabernet Sauvignon” and “Sangiovese” decreased linearly with increasing temperature. Our results parallel this decreasing trend in SSC when raspberries are exposed to higher temperatures. However, other studies have found the opposite trend in grapevines, where SSC increases with higher temperatures. For instance, Xyrafis et al. [53] evaluated the sugar content in grapes from 1987 to 2019 and found an increase in sugars of 2 °Brix. Likewise, increased sugars due to a shift in grape berry metabolism have been reported under high-temperature stress [54]. Another study in grapevines evaluated the effect of different temperatures applied at the *veraison* stage on total soluble solids [55]. Their findings indicated an increase in soluble solids in temperature treatments of 30 °C/15 °C compared to 22 °C/15 °C. A recent investigation by Rienth et al. [56] determined that an increase of 2–4 °C at different phenological stages consistently increased the SSC in three grape varieties during two seasons.

In blueberries, the SSC was negatively related to a thermal increase of 3.3 °C above the control temperature, imposed on blueberry plants [28]. In their study, SSC decreased by 13.4% in those blueberries from actively heated chambers, however, no changes in titratable acidity were evident. This would suggest a negative impact of increasing temperatures due to climate change on perceived consumer flavor since the sugar/acidity ratio might be changing.

A plausible explanation for this decrease in sugar content under high temperatures has been correlated with downregulation in the import and accumulation of hexoses [56]. In our study, the central orchard obtained higher SSC values than the southern orchard; however, when the heat treatment was imposed on the plants, the raspberries showed lower sugar content. In addition, Dominguez et al. [57], in their work on grapes, showed how high temperatures affect the ripening process, leading to increased pH mainly via malic acid degradation and more sugar content. These changes raised the alcohol levels and wine pH while decreasing acidity.

Berry acidity is another important quality parameter likely to decline in response to warming [58]. In grapes, the increase in temperature is expected to lower the acidity and increase the sugar content, resulting in unbalanced grapes with higher alcohol content, specifically ethanol, and deprived of freshness and aromatic complexity [59,60]. In our work, the sugar content of the heated raspberries decreased, and the acidity remained mostly unchanged or was not significantly different from the controls, similar to what has been reported by Alaba et al. [28] in blueberries. This reduction in sugar content and the unchanged acidity in raspberries under heat treatment should predict a shift in flavor towards more sour-tasting fruits. Interestingly, the sensory panel did not perceive this change in flavor in any of the two orchards (Figure 4). This could be explained by the fact that the number of raspberry fruits tasted by each panelist (3–4) may have been insufficient to detect differences as the fruit-to-fruit variability was high. Also, panelists were not provided any reference solution standards, which could have standardized their use of the scale.

Our results suggest that the heat treatment led to softer fruits, indicating potential challenges for postharvest life under future climate change scenarios. Interestingly, while the central orchard exhibited higher SSC than the southern orchard, the heat treatment resulted in reduced sugar content, a trend consistent with other studies on different fruit species. Despite these changes, the sensory panel did not detect significant flavor differences, possibly due to high fruit-to-fruit variability. Overall, the findings suggest that while increased temperatures may enhance certain growth parameters, they could adversely affect fruit quality, highlighting the need for adaptive strategies to cope with future climate conditions.

### 4.2. Antioxidant Properties

Red raspberries are widely in demand due to their distinct phytochemical profile, and high levels of polyphenols have been associated with human health benefits [61,62]. These components have a key effect in decreasing blood cholesterol, tumors, mutagens, and cardiovascular disorders [63]. Raspberries and blackberries are known for being good sources of polyphenolic compounds, such as anthocyanins, which have antioxidant capacity [64]. The concentration of these polyphenolic compounds can be induced by many factors: species, cultivar, ripening stage, soil, and climate conditions.

In the present study, fruit from the southern orchard exhibited higher antioxidant capacity than those from the central orchard. Total polyphenols content (TPC) varied: in 2023, the central orchard had higher TPC, while in 2024, the southern orchard led. Heating chambers generally reduced TPC in both orchards. Flavonoid content showed no differences in the first season. However, it varied in the second, with control raspberries in the central orchard and heated raspberries in the southern orchard having higher contents. Anthocyanin content mirrored TPC trends, with heating chambers reducing levels in the first season but increasing them in the southern orchard in the second. Statistical analysis revealed that orchard location consistently affected antioxidant capacity, while heat treatment impacted flavonoid content across both seasons. The interaction of orchard location and heat treatment significantly influenced TPC only in 2024 and flavonoid content in both seasons. Sugars play a crucial role as carbon skeletons in the production of phenylalanine, which is a key precursor in the anthocyanin pathway, indicating a direct relationship between anthocyanin levels and sugar content, as has been demonstrated in grape skin [65,66]. Our study shows no significant variations in sugars, so the decrease in anthocyanins observed in the heat chambers would not be due to sugar metabolism. On the other hand, the ORAC differences between seasons were unrelated to polyphenols, flavonoids, anthocyanins, or vitamin C; other molecules are likely associated with the differences in antioxidant capacity between the two seasons.

In raspberries, a study carried out by Freeman et al. [67] found that fruit from primocane raspberries grown in a hot, dry climate produced a higher antioxidant and polyphenolic compound content than berries grown in other climates. Similar results were found in strawberries, when grown at 30 °C, presented higher amounts of TPC, total flavonoid content, and anthocyanins than those grown at 25 °C [68]. In grapevines, Lecourieux et al. [69] evaluated a heat treatment (>8 °C than the control) in *Vitis vinifera* L. cv. “Cabernet Sauvignon” at three phenological stages and showed that when the heat treatment was applied during the green stage, *veraison* was significantly delayed in response to the late accumulation of sugars, and total anthocyanin decreased by 50% in *veraison* in the heated treatment.

Anthocyanin concentration can be modulated through degradation as a consequence of the action of different enzymes such as laccases, polyphenol oxidases, class III peroxidases, and β-glucosidases, where the heat treatment enhances the expression of genes impacting in polymerization rate of various polyphenolic compounds [69]. These authors also mention that these findings are consistent with previous work applying high temperatures (>30 °C) to fruits, resulting in anthocyanin degradation and inhibition of its accumulation, which may be subject to variety and type of anthocyanin. This negative relationship between anthocyanin accumulation and high temperature is also confirmed by Arrizabalaga et al. [70], where heated grapevines two weeks after mid-*veraison* significantly reduced their anthocyanins concentration, and at the same concentration of total soluble solids the anthocyanin concentration was lower at 28 °C/18 °C than 24 °C/14 °C, indicating a decoupling effect of elevated temperature during berry ripening. Therefore, the heat treatments reduced total polyphenols and anthocyanins observed in the first season, echoing the results of Lecourieux et al. [69] and Arrizabalaga et al. [70], who reported anthocyanin degradation at elevated temperatures.

The loss of nutrient components has been reported as part of the consequences of climate change [27]. Total polyphenols, anthocyanins, and vitamin C decreased in this study. A likely explanation for this shift in metabolism has been hypothesized in a meta-analysis considering several species, nutrients, and plant tissues [71]. These authors indicate that stressed tissues under climate change favor the upregulation of secondary metabolites related to anti-stress and survival functions. They also hypothesized that the impact of the change in environmental cues will be different among tissues, for instance, more nutrients will be translocated to roots instead of leaves to improve the intake of resources from the soil. Vitamin C has a significant role in photosynthesis as an enzyme cofactor and redox buffer [72]; considering that plants under heat stress might be downregulating primary metabolism such as photosynthesis, probably vitamin C is not produced at a normal rate.

## 5. Conclusions

Climate change and the concomitant temperature increase significantly affect red raspberry quality’s key attributes. A reduction in fruit firmness and soluble solids, along with unchanged fruit acidity levels, were observed, key aspects that can lead to a reduced market opportunity for the raspberries caused by their short postharvest life. The increased temperatures caused a shift in the secondary metabolites of raspberries, affecting antioxidant capacity, total polyphenolic compounds content, flavonoids, and vitamin C. Some of these sharply decreased, suggesting a potential dilution of nutrients of these essential compounds for human health in the future. At the sensory level, a related negative effect on fruit color was observed; however, sensory analyses with a larger sample size would be more appropriate to elucidate if consumers perceive quality attributes.

## Figures and Tables

**Figure 1 foods-14-01201-f001:**
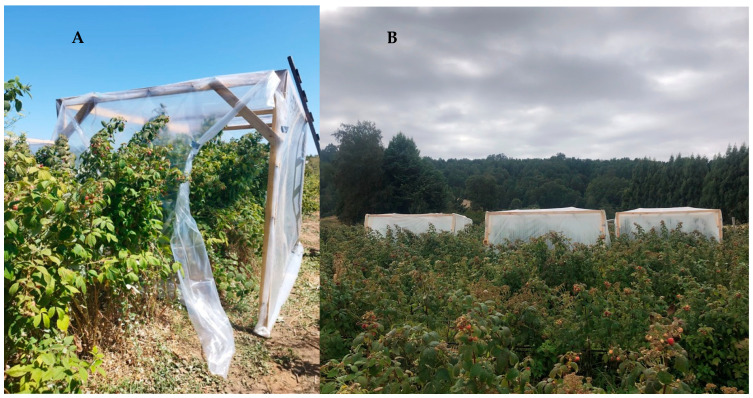
Heating chambers installed in raspberry orchards (cv. Heritage) in two contrasting agroclimatic locations in Chile. (**A**) High background temperature orchard in Central Chile, and (**B**) low background temperature orchard in Southern Chile.

**Figure 2 foods-14-01201-f002:**
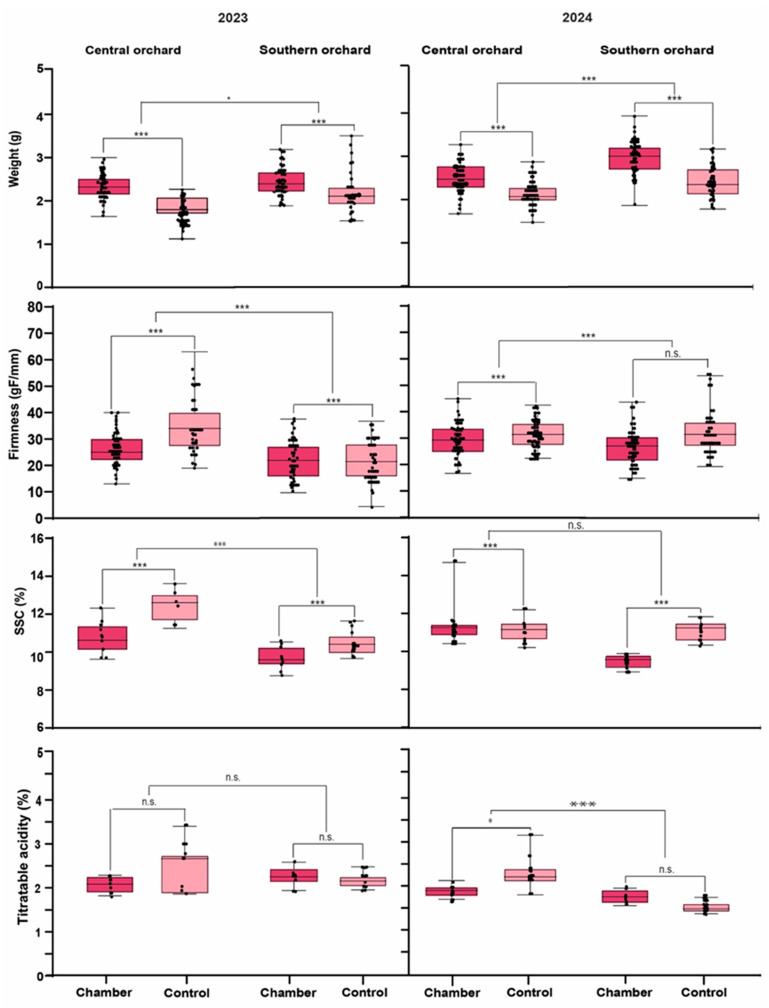
Fruit quality parameters for raspberries cv. “Heritage” for seasons 2023 and 2024. Dark pink box plots represent the heating chambers, while light pink box plots correspond to the control treatment. The boxes on the left side of each graph correspond to the data obtained from the central orchard, while those on the right side correspond to the southern orchard. The results of statistical comparisons between treatments and between orchards are expressed as asterisks, * *p* ≤ 0.05; *** *p* ≤ 0.001, n.s. not significant.

**Figure 3 foods-14-01201-f003:**
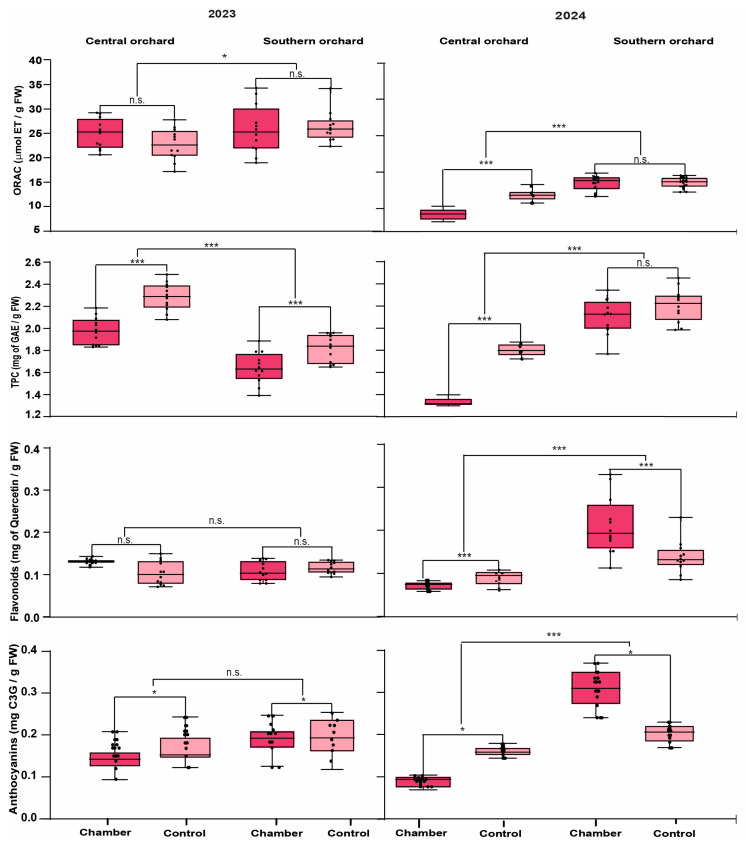
Antioxidant parameters for raspberries cv. “Heritage” for seasons 2023 and 2024. Dark pink box plots represent the heating chambers, while light pink box plots correspond to the control treatment. The boxes on the left side of each graph correspond to the data obtained from the central orchard, while those on the right side correspond to the southern orchard. The results of statistical comparisons between treatments and between orchards are expressed as asterisks, * *p* ≤ 0.05; *** *p* ≤ 0.001, n.s. not significant. ET: Trolox equivalents; TPC: total polyphenols content; GAE: gallic acid equivalents; C3G: cyanidin-3-glucoside.

**Figure 4 foods-14-01201-f004:**
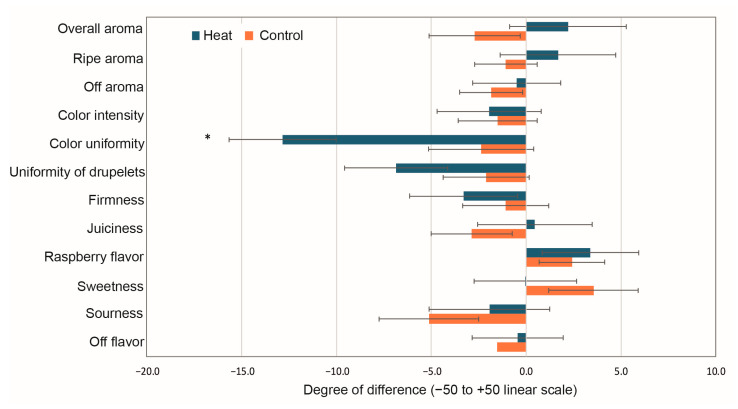
Fruit sensory parameters for raspberries cv. “Heritage” from the central for season 2024. The blue bars represent the heated fruit while the red bars correspond to the control treatment. Each bar is the mean rating (and standard error) of the difference between the coded sample (heat or unheated control) and the reference (unheated control). * Indicates significant difference between the heat-treated fruit and the control at *p* ≤ 0.05 using Fisher’s Least Significant Difference for multiple comparisons test.

**Table 1 foods-14-01201-t001:** Average temperature for maximum, minimum, and relative humidity conditions for the central and southern orchards during 10–12 days for seasons, 2023 and 2024.

Year	2023				2024			
Orchard	Central		Southern		Central		Southern	
Treatment	Control	Chamber	Control	Chamber	Control	Chamber	Control	Chamber
Average T° (°C)	18.91	20.45	13.16	17.55	18.97	22.63	12.30	16.25
Max T° (°C)	35.59	38.55	28.73	37.03	33.5	46.59	28.82	36.01
Min T° (°C)	4.96	7.47	1.96	6.63	5.42	5.72	−2.31	1.40
Relative Humidity (%)	66.41	66.07	83.07	84.53	74.19	70.97	86.65	82.92

**Table 2 foods-14-01201-t002:** *p*-values for fruit quality parameters in raspberries cv. “Heritage” for seasons 2023 and 2024, from central and southern orchards.

*p*—Value ^1^					
Quality Parameters					
Year	Effect	Weight (g)	Firmness (gF/mm)	SSC (%)	TA (%)
2023	Orchard	0.0109 *	1.55 × 10^−5^ ***	9.92 × 10^−10^ ***	n.s.
	Heat Treatment	1.01 × 10^−11^ ***	1.51 × 10^−11^ ***	2.30 × 10^−7^ ***	n.s.
	Orchard × Heat Treatment	0.0158 *	n.s.	0.0206 *	0.00139 **
2024	Orchard	1.48 × 10^−8^ ***	0.000386 ***	n.s.	4.64 × 10^−8^ ***
	Heat Treatment	<2 × 10^−16^ ***	4.62 × 10^−6^ ***	4.61 × 10^−10^ ***	0.0375 *
	Orchard × Heat Treatment	0.0329 *	n.s.	0.0128 *	2.36 × 10^−6^ ***

^1^ Statistically significant interactions or main effects within columns are indicated by asterisks: * *p* ≤ 0.05; ** *p* ≤ 0.01, *** *p* ≤ 0.001. Non-significant interactions or effects are indicated by n.s.

**Table 3 foods-14-01201-t003:** *p*-values of antioxidant parameters for raspberries cv. “Heritage” for seasons 2023 and 2024 from central and southern orchards.

	*p*—Value ^1^				
	Antioxidant Parameters				
Year	Effect	ORAC (μmol ET/g FW)	TPC (mg GAE/g FW)	Flavonoids (mg de Quercetin/g FW)	Anthocyanins (mg C3G/g FW)
2023	Orchard	0.0393 *	6.88 × 10^−14^ ***	n.s.	n.s.
	Heat Treatment	n.s.	5.42 × 10^−8^ ***	n.s.	0.0367 *
	Orchard × Heat Treatment	n.s.	n.s.	0.00156 **	n.s.
2024	Orchard	<2 × 10^−16^ ***	<2 × 10^−16^ ***	1.74 × 10^−14^ ***	<2 × 10^−16^ ***
	Heat Treatment	3.61 × 10^−11^ ***	3.10 × 10^−14^ ***	n.s	0.0164 *
	Orchard × Heat Treatment	2.24 × 10^−10^ ***	7.68 × 10^−10^ ***	9.15 × 10^−7^ ***	<2 × 10^−16^ ***

^1^ Statistically significant interactions or main effects within columns are indicated by asterisks: * *p* ≤ 0.05; ** *p* ≤ 0.01, *** *p* ≤ 0.001. Non-significant interactions or effects are indicated by n.s.

**Table 4 foods-14-01201-t004:** Vitamin C content of raspberries cv. “Heritage” for seasons 2023 and 2024.

Vitamin C (mg/100 g FW)				
Year	Orchard		Temperature	
2023	Central	24.04 a*	Heat	16.77 a
	Southern	16.81 a	Control	24.08 a
2024	Central	24.73 a	Heat	17.26 b
	Southern	17.99 b	Control	25.47 a

* Different letters within columns represent significant differences of each factor individually at a *p* ≤ 0.05.

**Table 5 foods-14-01201-t005:** *p*-values of vitamin C of raspberries cv. “Heritage” for seasons 2023 and 2024 from central and southern orchards.

*p*—Value ^1^		
Vitamin C (mg/100 g FW)		
Year	Effect	Vitamin C (mg/100 g FW)
2023	Orchard	n.s.
	Heat Treatment	n.s.
	Orchard × Heat Treatment	n.s.
2024	Orchard	0.0354 *
	Heat Treatment	0.0151 *
	Orchard × Heat Treatment	n.s.

^1^ Statistically significant interactions or main effects within columns are indicated by asterisks: * *p* ≤ 0.05;. Non-significant interactions or effects are indicated by n.s.

## Data Availability

The original contributions presented in the study are included in the article/Appendix A, further inquiries can be directed to the corresponding author.

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
