# Peer review of "Increased Temperature Effects During Fruit Growth and Maturation on the Fruit Quality, Sensory and Antioxidant Properties of Raspberry (Rubus idaeus L.) cv. Heritage"

_foods, 2025, doi:10.3390/foods14071201_

Round 1

Reviewer 1 Report

Comments and Suggestions for Authors

This is a novel study that discusses the impact of rising temperatures on the quality of red raspberries, shedding light on the global warming issue. However, the manuscript contains numerous careless errors that require careful check. Additionally, there are many unclear sections that need further revision to meet publication standards.

Line 19-23: Please condense this paragraph.
Line 70: Why is the impact of soilless cultivation on red raspberries discussed? This article only focuses on the effects of soil on red raspberries.
Line 141: Pay attention to the reference format; "according to" should be followed by the author's name. Please also correct this error in other place.
Line 158: What is the specific method used by the FirmPro equipment to detect raspberry fruit? Please provide a reference or describe the method, as parameters can affect the results.
Line 166: How was the endpoint of pH 8.2 determined? Was a pH meter used? However, 2 ml of juice is too small for accurate pH measurement with a pH meter. Was pH paper used? Even then, 2 ml of juice is insufficient for pH paper. Did you use a pH indicator instead?
Line 176: Filter paper filtration may lead to sample loss. Did you use solvent to rinse the filter paper?
Line 181: What did you use to hold the samples and reagents? A black 96-well plate? Please add it into the manuscript
Line 182: What is the volume of PBS used?
Line 183: Some reagent information is unclear, such as APPH and fluorescein. Which company were they purchased from? Please check other parts of the manuscript.

Line 185: I have conducted ORAC experiments before. I remember that ORAC experiments only require measuring fluorescence intensity every 2 min. Why did you measure antioxidant capacity every minute? What is this metric? I also checked the reference you cited and found that it does not mention measuring antioxidant capacity. Please review the ORAC experimental method.

Line 195: TPC protocol?

Line 204: What was used to dilute the samples? What was the sample volume?

Line 215: How much HCl was used?

Line 227: References are missing. What is the model of the HPLC, and which company was it purchased from?

Line 231: 10.000 × g??

Line 259-261: "This test is used to determine if there are differences between two samples (typically the test sample and a control) and estimate the size of such difference." I believe this sentence can be removed.

Line 351: The antioxidant section can be separated from the fruit quality section.

Line 352: I noticed significant differences in ORAC results between 2023 and 2024. Why is this not emphasized in the manuscript?

Line 353-365: Please add units to the experimental values. Review the entire manuscript and add missing units where necessary.

Line 451-456: Please avoid rewriting the results in the discussion section. Instead, discuss the reasons for the observed differences and provide explanations for the results.

Line 469: Why do these different phenomena occur? Do you have any explanations?

Line 477-490: Do not simply list research findings. Compare them with your results or use their findings to explain your results.

Line 504-507: Grammar error.

Table 4 and Table 5: The line numbers overlap with the table content. Will this affect the formatting? Please correct this.

Reviewer 2 Report

Comments and Suggestions for Authors

The author examined how increased temperature affects the quality and flavor of red raspberry fruits. The findings reveal that a 4-degree rise in ambient temperature results in larger yet softer raspberries with a reduced soluble solids content compared to the control group. This study is well-designed, supported by substantial and detailed data. 
However, there are several areas where improvement or supplementation by the author is necessary: 
(1)  In Table 1, the author should change the commas to decimal points in the presented data.
(2) It should also be clarified whether the temperature increase caused red raspberries to mature earlier, and if so, the approximate number of days by which maturity was accelerated. 
(3) The author notes that "harvest occurs when the raspberry reaches the red ripe stage," implying similar harvest colors. The author should explain whether this is the reason for the lack of difference in anthocyanin content and is recommended to provide comparative photos of the harvested fruits. 

Reviewer 3 Report

Comments and Suggestions for Authors

The manuscript explores the impact of increased temperatures on the quality, sensory, and antioxidant properties of raspberry (Rubus idaeus L.) cv. Heritage. The study is relevant in the context of climate change and its effects on fruit crops. The research design appears well-structured, including a clear methodology and statistical analysis. However, there are several issues that need attention before it can be accepted.

Why did you decide to use ORAC as an assay for antioxidant capacity

Which was the standard in the TPC?

Detail the software used for statistical purposes

Some claims in the discussion lack strong supporting references, particularly regarding the mechanisms by which temperature affects antioxidant content and fruit quality.

The study does not sufficiently discuss how well these conditions mimic real climate change scenarios (e.g., fluctuations in temperature, humidity, and radiation).

The choice of two locations is appropriate, but the manuscript does not clearly describe the potential confounding effects of other environmental factors, such as soil composition or water availability

The manuscript does not clearly specify how many samples were used for each analysis of fruit quality, antioxidants, vitamin C, and sensory evaluation.  It mentions that "Twenty raspberry fruit per treatment were used for quality assessments." However, it is unclear whether this refers to each year, each location, or how the samples were divided across replicates.

About antioxidant analysis the number of samples per treatment and the number of replicates are not specified.  It only states that extractions were performed using 2 g of sample, but it does not clarify how many biological or technical replicates were included.

About vitamin C analysis it mentions using "approximately 10 g of raspberry frozen samples," but does not specify how many samples per treatment or the number of replicates.

About sensory analysis It states that "Ten panelists participated in three sessions." However, it does not specify how many samples were evaluated in total or whether any repeatability analysis was performed among panelists.

The author should provided details such as number of fruits per treatment and location used in each analysis; number of biological and technical replicates for each test (ORAC, TPC, vitamin C, etc.) and whether sensory analyses were replicated to confirm the consistency of results.

Comments on the Quality of English Language

The English in the manuscript is generally understandable, but it has several issues related to grammar, clarity, word choice, and sentence structure.  Example: "The analysis of each sample was performed in triplicate." → While correct, a better phrasing would be "Each sample was analyzed in triplicate."

Round 2

Reviewer 1 Report

Comments and Suggestions for Authors

There are still some careless mistakes, such as 6 x 2 x 2.10 m, which I think should be m³, or 6 m x 2 m x 2.10 m? Besides, why is it 2.10 instead of 2.1? Or why not 6.0?

Line 110: Provide more information about transparent polyethylene plastic film.

Line 178: acetone/water/acetic acid, v/v/v?

Line 204-205: Is it necessary to homogenize twice in a short time?
Please give the speed in terms of g values rather than rpm.

There are many reagents whose purchase sources are unclear, such as quercetin and cyanidin-3-glucoside.

How is the difference between the sample and the control group evaluated in sensory analysis? What kind of difference is considered -50? What kind of difference is considered +50?
For hardness, what kind of difference is negative? What kind of difference is positive? For example, what is the hardness result of a sample softer than the control group? This information should be told to readers.
Have you calculated how much the dry weight of the fruit has increased? Because it can be seen that the weight of the fruit has increased but the water content has also increased. What if the weight increase of the fruit is all from water? Can we conclude that the heating system has increased the photosynthesis rate?
The increase in water content of the fruit may cause it to soften, not necessarily because of the increase in calcium content. Have you tested the calcium content of the fruit?

Line 578: what is other molecules?

Line 579-581: Delete this sentence

Line 628: What is more robust sensory analyses? What is the difference between robust sensory analyses and sensory analyses in your study
Please discuss more about the impact and implications of temperature rise on agriculture.
